# Emotional Warmth and Rejection Parenting Styles of Grandparents/Great Grandparents and the Social–Emotional Development of Grandchildren/Great Grandchildren

**DOI:** 10.3390/ijerph20021568

**Published:** 2023-01-14

**Authors:** Yang He, Chengfang Liu, Renfu Luo

**Affiliations:** School of Advanced Agricultural Sciences, Peking University, Beijing 100871, China

**Keywords:** parenting styles, social-emotional development, mediation analysis

## Abstract

Parenting styles are crucial in the process of forming social emotions in children. They are also vital for creating effective family policies in order to improve a child’s early development. As such, it is important to acknowledge the enduring association of parenting styles across generations, as well as their impact on early child development. In this study, the question as to whether the warm and hostile parenting styles of a parent/grandparent mediate the relationships between the emotional warmth and rejection parenting styles of a grandparent/great grandparent, as well as the subsequent social–emotional development of a grandson/great grandson and/or a granddaughter/great granddaughter, was examined. Cross-sectional assessment data from 194 primary caregivers of children between 6 and 36 months were analyzed using mediation analyses. In addition, moderated mediation models were used to test heterogeneity effects. This study found evidence that the warm and hostile parenting styles of a parent/grandparent mediated the associations between the emotional warmth and rejection parenting styles of a grandparent/great grandparent, as well as the subsequent socio-emotional development of a grandchild/great grandchild. Parents/grandparents tend to use a warm parenting style when the child is a boy, thereby resulting in fewer socio-emotional problems. This study provides empirical evidence for the purposes of preventive services to improve caregivers’ parenting styles in the early stages of a child’s development. Researchers and family practitioners should continue to support families with intervention or therapeutic techniques in order to mitigate potential lasting consequences.

## 1. Introduction

How much do the parenting styles of primary caregivers depend on their perceived parenting styles? In addition, how do parenting styles influence human capital formation? These questions are of long-standing interest in the social sciences, partly due to the fact that the answers reveal something about how human capital and inequality are transmitted across generations.

Differences in human capital among individuals begin to emerge early in life and early life experiences have long-term effects on how people develop. Human capital is widely defined as the possession of certain skills (both cognitive and noncognitive), as well as certain capabilities, including in relation to health or social functioning [1]. The formation and development of human capital is a gradual and cumulative process, where achievement gaps emerge at age one for children who are from families of different socioeconomic levels [2]. Children who acquire better cognitive and noncognitive abilities not only become more productive adults [3,4], but mounting research also links the educational attainment and parenting styles of parents to the cognitive and noncognitive development of children of all age groups [5]. Furthermore, increasing social inequality reduces intergenerational mobility and leads to social entrenchment [6].

Depending on the extent to which parents support or demand their children, American developmental psychologist Diana Baumrind laid out four types of parenting styles that have been widely referred to in the literature: authoritative, authoritarian, permissive, and neglectful [7]. This categorization provides us with a comprehensive insight into the parenting styles of caregivers. However, further studies in later years dug deeper into the characteristics of these different styles and proposed relative dimensions of parenting styles in order to aid with more targeted and specific policy recommendations. For instance, Arrindell et al. (1986) summarized three types of parenting styles in order to measure perceived parenting styles in childhood, including emotional warmth, rejection, and overprotection [8]. Emotional warmth has been described as “the quality of loving relationships between parents and their children and physical and verbal behaviors parents use to express those feelings” [9]. Rejection refers to disapproval, criticism, arbitrary censure, or punishment from parents [10].

Moreover, empirical studies have shown evidence of parenting style transmission between generations. Despite measuring parenting styles at different ages with different methods, many studies show that positive and negative parenting styles are transmissible to the next generation [11,12,13,14]. Early exposure to harsh or abusive parenting is probably the most consistent predictor of the subsequent adoption of a coercive parenting style [11].

There is an extensive body of literature that recognizes the importance of parenting styles, which play a crucial role in forming an individual’s lifelong psychological and physiological well-being [15,16,17]. Many economic and psychological studies further concentrate on the intergenerational transmission of parenting styles and how parenting styles influence human capital formation, especially during the early stages of life. For example, a parent’s appropriate reactions to the emotional needs of children, their tone and expressions with direction, their encouragement and love rather than commanding and/or coercive behavior, support their child’s social development [17]. In particular, in one study, decreased spanking and controlling behaviors from parents were associated with improvements in the children’s academic and behavioral skills [15].

Emerging evidence also indicates that a child’s socio-emotional competencies provide the foundation for future academic success and contribute to a prosperous and harmonious community. For example, children with better executive functions performed better in early math, language, and literacy development [18,19]. In addition, by developing critical social–emotional skills—such as building relationships with others, creating self-awareness, regulating and controlling emotions, and gaining independence—socially competent children are better adjusted and able to accept diversity, change, and new forms of learning. In addition, they will have increased opportunities to participate in school, at home, at work, and elsewhere throughout their lives [20,21]. Francesconi and Heckman (2016) documented that the interventions aimed at a child’s social–emotional skills are more effective than those targeting cognitive skills. This is due to the fact that social–emotional skills have a greater malleability than cognitive skills, which, in turn, usually only become stable after certain ages [22]. In contrast with cognitive skills, social–emotional skills have more lasting effects regarding lifetime welfare and thus deserve more attention.

However, many studies show that children in rural China struggle with social–emotional delays. These studies defined a delay by a social–emotional score of one or more standard deviations below the mean of a reference population’s expected developmental trajectory [23]. According to new estimates, 53% of children aged 0–3 years in four subpopulations of rural China have not attained their social–emotional developmental potential [24]. By utilizing longitudinal panel data from 1245 children in rural Western China, Wang et al. (2022) found that 30% of their sample children were persistently delayed in regard to their social–emotional development; in addition, 31% experienced deteriorating social–emotional development [24].

Although existing studies have provided good evidence with regard to intergenerational parenting styles, as well as the importance of caregivers’ parenting styles on children’s early development, there are still several limitations. First, most of the research was conducted in Western countries, where co-residence between grandchildren and grandparents is rare. Experiences from China are worth studying for the purposes of all the other countries that possess a culture of skipped-generation households. Second, some studies suggested that parenting styles are associated with a young child’s social–emotional development in the two-generation model. However, no dataset has provided information on the parenting practices of great grandparents. This study is the first to investigate the relationship between the emotional warmth and rejection parenting style of grandparents/great grandparents and the social–emotional development of their grandchildren/great grandchildren, as well as the first to attempt to understand this through the underlying mechanisms that operate within rural China.

Rural communities in China have become an ideal context in which to study the intergenerational transmission of parenting styles, its association with the development of later generations, and the role that gender inequality plays in these same associations. First, China’s population is aging faster than almost any other country in modern history, resulting in “long years of shared lives” between generations, especially in rural regions. Broad demographic trends and local economic conditions are creating more diversity in the ability of pastoral caregivers to care for their descendants and in the capacity of communities to support their elderly residents as family caregivers. Second, intergenerational solidarity in Asian cultures has created a sizable number of skipped-generation households, which is indicated by the co-residence between grandchildren and grandparents. Grandparenting is unavoidably becoming an efficient way to manage time constraints within families. Third, according to the theory of brain development, the environment (in addition to genes) plays a major role in determining the formation of a person’s development of their abilities [25]. It must be noted that patriarchy is a common phenomenon in rural China. In this paper, the aim is to find the extent to which son preference as an environmental influence in patriarchal culture affects the association between the parenting style of caregivers and the socio-emotional development of young children.

In this study, newly collected multigenerational data from rural China are used. The respondents are the primary caregivers of children 0–3 years of age. Depending on the identity of the primary caregiver, the intergenerational transmission of parenting styles implies three or four generations in this study, hereby collectively called “intergenerational transmission.” In order to better illustrate the intergenerational association model, we anchor the respondents of our field survey as G2 and extend the related associations thereafter. The first scenario demonstrates the association between the parenting styles of the grandparents, the parenting styles of the parents, and the neurodevelopment of the young child. In this scenario, parents (G2) are the primary caregivers of the young child (G3). In the second scenario, we examine the association between the parenting styles of the great grandparents, the parenting styles of the grandparents, and the neurodevelopment of the young child. In this scenario, due to the absence of parents, grandparents (G2) are the primary caregivers of the young child (G4).

In this study, we aim to reveal the intergenerational associations across three or four generations, as well as the mechanisms by which they operate. In order to achieve this objective, we try to answer the following questions: To what extent is the emotional warmth and rejection parenting style of grandparents/great grandparents (G1) associated with the social and emotional development of their grandchild/great grandchild (G3/G4)? What is the mediating mechanism underlying intergenerational associations? Does gender matter when analyzing the potential consequences of intergenerational associations?

## 2. Theoretical Background and Hypotheses

A host of studies have shed light on the underlying mechanism of the transmission of parenting styles through generations. One interpretation of the connection between the parenting styles of the two generations is that it simply reflects observational or experiential learning (e.g., [26]). Other scholars have suggested that parenting styles are transmitted to the next generation indirectly through generous support from a romantic partner [27], educational attainment [12], externalization behavior [28], and mental health conditions [13].

**Hypothesis** **1.**
*The emotional warmth and rejection parenting styles of G1 are positively associated with the warmth and rejection parenting styles of G2 (path a in Figure 1).*


A preschool-aged child’s social–emotional development can be influenced in various ways, one of which is through their primary caregivers’ parenting styles. Children raised in supportive or authoritative households possessed a greater resilience and psychological competence, whereas authoritarian or permissive parenting may be deleterious [29,30,31,32,33]. For instance, in two studies conducted in Pakistan and Spain, the authoritarian parenting style was positively associated with internalizing and externalizing problems among the sample children, whereas the permissive parenting style was positively associated with internalizing problems [29,30,34]. In another Finnish longitudinal sample, maternal psychological control elevated negative emotions in young children, especially among temperamentally vulnerable children [34]. Furthermore, Xing et al. (2011) found that aggressive parenting predicted externalization problem behaviors in girls but not in boys [33].

Based on the above literature, in this study, we propose the following hypotheses regarding the association between emotional warmth and rejection parenting styles of the primary caregivers in relation to the social–emotional development of later generations:

**Hypothesis** **2a.**
*The warm parenting styles of G2 as primary caregivers are positively associated with the social–emotional competencies of G3/G4 and are negatively related to the social–emotional problems of G3/G4 (path b in Figure 1).*


**Hypothesis** **2b.**
*The hostile parenting styles of G2 as primary caregivers are positively associated with the social–emotional problems of G3/G4 and are negatively related to the social–emotional competencies of G3/G4 (path b in Figure 1).*


As parenting styles can transmit to the next generation, we further proposed a hypothesis regarding the mediation effect of G2’s parenting style in the context of intergenerational associations:

**Hypothesis** **3.**
*G2’s warm and hostile parenting style mediates the relationship between the emotional warmth and rejection parenting style of G1, as well as the social–emotional development of G3/G4.*


Various factors influence parenting styles; one of them is the child’s gender. Traditionally, it was believed that parents would devote a different amount of time and resources to their sons and daughters. Previous studies have revealed that fathers generally exhibit more divergent parenting behaviors for their sons and daughters, more so than is the case for mothers [35]. There were also discussions regarding the moderated effect of gender within the study of parenting styles, one of which saw a substantial difference in the authoritative parenting style of mothers with sons and daughters, but no significant difference in the authoritarian parenting style of mothers between sons and daughters [36].

On this basis we therefore proposed Hypothesis 4:

**Hypothesis** **4.**
*The gender of G3/G4 moderates the mediation effect of the warm and hostile parenting style of G2 (path c in Figure 1).*


## 3. Materials and Methods

### 3.1. Study Participants

The research team drew participants from an investigation conducted in three townships. These townships were all in one county, which was randomly selected from the 21 counties that have been nationally designated as the poverty-stricken counties within the Jiangxi province [37]. Three sample townships were also randomly selected from a list of all the townships in the sample county. We randomly selected 200 children aged 6–24 months living in the sample townships. This was based on the information that was obtained from a list of the registered births from local healthcare officials in each of the sample townships. Of the 200 children that were to be included within this study, 6 possessed rural household registration, but did not actually live in the sample village. As such, they were not considered in our target sample. Therefore, in the end, this study involved 194 children.

The Peking University Institutional Review Board (PU IRB), Beijing, China, approved the ethical assessment of the study (No. IRB00001052-19132). The research team explained the purpose of the study and obtained verbal informed consent from the caregivers of all children.

### 3.2. Data Collection

The data used in this study were collected from sample households over a 3-week investigation. The survey team collected four types of information: (1) the parenting styles of the primary caregiver’s parents, measured by the Egna Minnen av Barndoms Uppfostran inventory (EMBU), which consists of three subscales: rejection, emotional warmth, and (over) protection; (2) the parenting styles of the primary caregivers, as measured by the parenting practice questionnaire (PPQ), which includes four subscales: warm, consistency, hostile, and hostility; (3) the social–emotional problems and competencies of young children, as measured by the brief infant–toddler social and emotional assessment (BITSEA); and (4) the household socio-demographic characteristics. Due to the fact that different periods of parenting styles were collected at the same time, the data thus used in this study possess an intertemporal feature that implies implicit temporal ordering.

The primary caregiver is the person responsible for the child’s daily care. We trained college students as the enumerators who were blind to the study hypotheses in order to better administer interviews with the primary caregivers of children in the sample households. Enumerators received thorough training for one week in order to ensure that they consistently understood the survey and could conduct it in a standardized fashion.

Through conducting the research, a series of measures were constructed in order to ensure quality data collection. First, the team carefully developed the research protocols and questionnaire at the proposal stage. Next, the pretesting around the sample areas was conducted and it was an invaluable component of this research. It allowed the team to identify questions that did not make sense to participants. In addition, it also allowed the team to investigate the problems with the questionnaire that may lead to biased answers. Additionally, crosschecking was conducted at the enumerator level, and the research team routinely conducted special investigations. Throughout each quality control stage, the research team focused on resolving anomalies and ensuring that concerns were recognized and addressed promptly.

### 3.3. Measures

(1)Parenting styles of great grandparents/grandparents

The Egna Minnen av Barndoms Uppfostran (EMBU) inventory is one of the most commonly used retrospective inventories. It is utilized in order to assess how individuals perceive their parents’ parenting styles separately. The original version of the EMBU consists of 81 items. Researchers from Australia, Denmark, Hungary, Italy, Netherlands, and other countries have also revised the EMBU and conducted cross-cultural research [8]. Yue (1993) conducted a pilot study examining the psychometric properties of the Chinese version in the late 1980s [38]. Given the time constraint during data collection, we adopted a short form (s-EMBU) version in this study consisting of 23 items. This short-form consists of three subscales: emotional warmth, rejection, and (over) protection. There are 7, 6, and 10 items in the abovementioned three subscales. A study conducted by Arrindell et al. (1999) examined the factorial, construct validity, and reliability of the s-EMBU among 1331 students from Italy, Hungary, Guatemala, and Greece. They also provided the scoring key, the instructions for filling out the form, and the recommended s-EMBU as a reliable and functional equivalent inventory to the 81-item previous EMBU [39]. Due to the fact that retrospective questions require long-term memories regarding their perceived parenting styles during childhood, this part of the questionnaire was intentionally placed after the survey, such that expectations arising from a measure of message recall did not bias other data collected from that participant.

Questions in the s-EMBU can be categorized into three subscales and are asked separately regarding experiences with the participant’s parents. Therefore, in this study, we used four subscales to capture the parenting styles of great grandparents/grandparents: the emotional warmth of both the parents separately, as well as emotional rejection of both the G1 parents separately. In this paper, we calculated the total score of the parents in the context of the emotional warmth and rejection subscales, and then standardized them in the analysis. The central concept of the emotionally warm parenting style subscale is: being affectionate, stimulating, and praising. This can be exemplified with questions such as “did you feel that warmth and tenderness existed between you and your parents?” and “do you think that your parents tried to make your adolescence stimulating, interesting and instructive?”. The rejection parenting style subscale is characterized by hostility, punishment, shaming, and rejection through criticism. Example questions of the rejection subscale include “do you feel it was difficult to approach your parents?”, “did it happen that your parents punished you, even for small offenses?”, etc.

According to Arrindell et al. (1999), all the items in the s-EMBU inventory were scored via the 4-point Likert-type scales (1 = never; 2 = yes, but seldom; 3 = yes, often; 4 = yes, always) [39]. A higher score indicates a higher level of emotional warmth or rejection experienced by the caregiver. Internal consistency and reliability were tested through Cronbach’s alpha. The results show that Cronbach’s alpha coefficient was 0.74~0.81 in relation to this study’s sample, which is good [40].

We calculated the combined scores and difference scores of G1’s parenting styles and used them as control variables. The combined score represents the overall parenting environment of the family. Furthermore, the difference score represents the difference of the parenting styles of the two caregivers (in this case, grandparents or great grandparents). Both of the two scores have a potential effect on the development of young children [41,42]. The problem is that the difference score may be correlated with the values used to compute them; further, they can also compound measurement errors due to regression to the mean [43]. This study followed the procedures that Solomon and Theiss (2008) outlined on how to address the above problem [44]. We first regressed the parents’ emotional warmth and rejection parenting styles onto the parenting styles of the next generation via separate analyses. The resulting standardized beta coefficients were used in order to weigh each independent variable. The predicted values of each parenting style score were used to calculate the difference between the parents.

Specifically, this study took the arithmetic mean scores of G1’s parenting styles when calculating the combined scores of the parents. When calculating the dissimilarity, we took, in this study, the absolute value of the differences but made the absolute value negative. Therefore, we obtained variables that indicate the score differences between the parents, in which 0 represented a complete similarity between the spouses. The resulting variables had the potential to range from −2.25 (spouses are entirely different) to 0 (spouses are completely similar).

(2)Parenting styles of grandparents/parents

The parenting practice questionnaire (PPQ) is a self-report questionnaire that is completed by the child’s primary caregiver in order to measure parenting styles. Parents from the United States, Australia, China, and Russia examined the psychometric characteristics of the questionnaire (Robinson, 1996), which has resulted in the questionnaire showing similar overall parenting styles across the four cultures.

The PPQ consists of four subscales: warm, consistency, hostile, and hostility. Thus, we established two subscales in this study: the warmth and hostility of the primary caregiver. The “warm” subscale in the PPQ refers to the degree to which the parents respond to their children in warm, encouraging ways and emphasize their child’s autonomy. The “hostile” subscale refers to how parents express rejection and behave as if they do not care about their child. The “hostile” and “warm” subscales of the PPQ, which are selected to be analyzed in this paper, include six items in each subscale, such as: when the child misbehaves or refuses to do what the parent wants them to do, then does the parent either scold, yell, threaten to punish, administer a spanking, slap, or hit. The response format of the items is constructed on a 5-point scale: 1 = never/hardly ever, 2 = seldom, 3 = sometimes, 4 = often, and 5 = always. Scores were averaged, and the higher scores reflected greater levels of that parenting style. The PPQ has also demonstrated satisfactory reliability scores, with Cronbach’s alpha coefficients of 0.59~0.78, thereby indicating that our sample’s internal consistency was acceptable [40].

The PPQ instrument evaluates current parenting styles based on the premise that the EMBU is typically used to measure people’s memories of upbringing. The “emotional warmth” and “rejection” subscales in the EMBU and the “warm” and “hostile” subscales in the PPQ measure similar parenting styles, respectively. Therefore, in this study, we used “emotional warmth” and “rejection” scores in the EMBU as the independent variables. In addition, the “warm” and “hostile” scores in the PPQ were established as dependent variables in order to better chart the transmission of parenting styles between the two generations.

(3)Social–emotional problems and the competencies of G3/G4

The 42-item brief infant–toddler social and emotional assessment (BITSEA) is a standardized norm-referenced instrument. It is designed to be completed by the child’s primary caregiver in order to measure a child’s social–emotional problems and competencies before they are three years old [45]. The questions in the BITSEA were drawn from the pool of 169-item infant–toddler social and emotional assessments (ITSEA, [45,46]). The clinical validity and reliability of the BITSEA have been verified in both psychiatric clinical samples of toddlers [47], as well as in randomly selected samples of young children [48]. The evaluator can complete the BITSEA questionnaire in about 5 to 7 min with at least a fourth- to sixth-grade reading level [45].

There are two critical components in the BITSEA that should be mentioned: social–emotional problems and social–emotional competencies. The “Problems” section includes items to measure externalization problems, internalization problems, problems of dysregulation, maladaptive behaviors, and the atypical behaviors of young children [48]. Sample items are such questions as: “is restless and cannot sit still?”, “hits, bites, or kicks you?”, and “does not make eye contact?”. The “Competencies” section measures social–emotional abilities, such as sustained attention, compliance, mastery motivation, prosocial peer relations, empathy, imitation/play skills, and social relatedness [48]. Sample items in this vein can be found in questions such as: “is affectionate with loved ones?”, “plays well with other children?”, and “can pay attention for a long time?” They are asked to score the items using 3-point scales (0 = not true/rarely, 1 = somewhat true/sometimes, 2 = very true/often). Scales were calculated as sums, which meant that the lower scores in the “Problem” subsection and the higher scores in the “Competence” subsection represented the better social–emotional conditions in which the child was currently situated in. The internal consistency reliability was acceptable across these subscales (Cronbach’s alpha: problem = 0.82 and competencies = 0.64).

(4)Confounding factors

We also collected data on the factors that could confound the intergenerational transmissions of parenting styles, which include three categories of variables: (1) child characteristics: gender (boy/girl), age in months (mean ± standard deviation), low birth weight (yes/no), and the number of siblings at home (mean ± standard deviation); (2) primary caregiver characteristics: age of grandparent/parent at the time of data collection and the decade of birth as dummy variables, education level (middle school or above = 1 and below middle school = 0), dissimilarity in scores of perceived emotional warmth/rejection experience with the parents in G1; and (3) household characteristics: whether the primary caregiver is the parent or the grandparent of the young child, whether the family is receiving subsistence allowances (yes/no). Within this study, we further controlled these confounding factors in the regressions in order to increase the estimation accuracy.

## 4. Empirical Strategy

First, sample characteristics are described and presented with means and standard deviations of the continuous variables and numbers (in percentages) of the categorical variables used in the analyses. Second, in the multivariate linear regression, the parenting styles of G1 (such as individual scores, total scores, and dissimilarity in the parenting styles scores of the grandfather/great grandfather and the grandmother/great grandmother) are dependent variables. In contrast, the parenting styles of G2 are conducted, as independent variables, in order to examine the intergenerational transmission of parenting styles in the multivariate linear regression under different circumstances.

After this, we tested our study hypotheses in two interlinked steps. First, we examined a simple mediation model in order to examine the associations between the parenting styles of G1, the social–emotional development of G3/G4, and the parenting styles of G2 as a mediator within the association. Second, we integrated the proposed moderator variable into the model and empirically tested the overall moderated mediation hypothesis. Further, statistical analyses were performed using Stata 15.1 software (StataCorp LLC, Texas, USA) and the significance level was set to *p* < 0.05 (two tailed).

### 4.1. Test of Mediation

In this step, we adopted the following model in order to identify the mediation effect of G2’s parenting style in the context of the relationship between the parenting style of G1 and the social–emotional development of G3/G4:(1)$$parenting2nd_{i}=\alpha_{1}+\beta_{1}parenting1st_{i}+\gamma X_{i}+u_{j}+\varepsilon_{i}$$
(2)$$socio_{i}=\alpha_{2}+\beta_{2}parenting1st_{i}+\beta_{3}parenting2nd_{i}+\gamma X_{i}+u_{j}+\varepsilon_{i}$$
where $parenting2nd_{i}$ is the parenting style of G2, the current primary caregiver of G3/G4; $parenting1st_{i}$ is the parenting style of G1; $socio_{i}$ is the measurement of the social–emotional problems and social–emotional competencies of G3/G4; $X_{i}$ refers to the covariates of socioeconomic characteristics, including the child’s gender, age in months, whether the child was born with low birth weight, the caregiver’s age, their decade of birth, the caregiver’s educational attainment, how many siblings are at home, etc.; $u_{j}$ is the village fixed effects in order to control for the unobserved heterogeneity at the village level; and $\varepsilon_{i}$ is the error term. On this note, we adjusted the standard errors to account for clustering at the village level.

$\beta_{1}$ indicates the indirect effect of the parenting style of G1 on the mediator; $\beta_{2}$ refers to the effect of the parenting style of G1 on the social–emotional development of G3/G4 adjusted for G2’s parenting styles; and $\beta_{3}$ refers to the effect of the social–emotional development of G3/G4 on G2’s parenting styles adjusted for the parenting styles of G1.

The hypothesis drawn from the above equations is that: when the effect of the independent variable on the dependent variable decreases by a nontrivial amount—but is not zero—and $\beta_{1}$ and $\beta_{3}$ are both significant, and $\beta_{2}$ is significant as well, then the mediation is partial. When the effect of the independent variable on the dependent variable decreases to zero, $\beta_{1}$ and $\beta_{3}$ are both significant, while $\beta_{2}$ is not significant, then this is complete mediation, according to [49]. When one of $\beta_{1}$ and $\beta_{3}$ is significant while the other is not, we conducted a Sobel test in order to determine if the mediation is partial or if there is no mediation.

In addition to Equations (1) and (2), the traditional mediating effect model is also required to test the causal relationships between the main predictors. In this case, this would be the parenting style of parents in G1, as well as the dependent variables, the social–emotional problems, and competencies of G3/G4. However, certain changes took place in the mediation effect model when more and more application and development was performed. MacKinnon et al. (2002) and Shrout et al. (2002) questioned the necessary condition of “*X* significantly affects *Y*” that is required by the traditional mediation model [50]. The reasons are as follows: (1) It is possible that several mediating paths exist in one mediating analysis concurrently. The overall effect will be concealed if the mediating effects are similar in size yet are presented in opposite directions. (2) When it possesses a more complicated mediating effect mechanism, *X’s* influence on *Y* may have other transmission channels in the causal chain, or be affected by competitive factors and random factors, such that the impact of *X* on *Y* will become smaller or perhaps even no longer significant. Therefore, a significant overall effect is not necessary for the purposes of considering the mediating effect.

Considering the fact that the main effect is hard to observe when mediating effects cancel out, we thus verified the indirect effects. We tested the significance of the mediating paths by using the bootstrap method based on a resampling with 1000 replications. There are several ways in which to construct bootstrap confidence intervals. Here, we included two types of 95% confidence intervals (CI) in order to test the statistical significance of the indirect effects through the mediator. The percentile interval is a first-order interval formed from quantiles of the bootstrap distribution without bias correction. In regard to a 95% confidence interval, the percentile confidence interval uses the 2.5% and 97.5% percentiles of the bootstrap estimates. The bias-corrected interval is a second-order accurate interval that corrects for bias in the distribution of bootstrap estimates. The bias is estimated as the difference between the statistics are calculated using all of the data and the mean value of the statistics across the bootstrap samples [51]. The indirect effect is considered statistically significant if the confidence interval does not contain zero.

### 4.2. Test of Moderated Mediation

Moderated mediation, i.e., conditional indirect effects, occurs when a moderator variable interacts with a mediator variable. The value of the indirect effect changes depending on the value of the moderator variable [49]. All the continuous variables were mean centered before the moderated mediation analysis was conducted [52]. In this study, we adopted the total effect moderation model posited by Edwards and Lambert (2007). This was performed by comprehensively presenting the mediation effect and the moderation effect in the same analytical framework, which, in turn, overcomes the shortcomings of the separated mediations and moderation analyses that were conducted in previous studies.
(3)$$parenting2nd_{i}=\alpha_{3}+\beta_{4}parenting1st_{i}+\beta_{5}gender_{i}\;+\beta_{6}parenting_{i}\times gender_{i}+\gamma X_{i}+u_{j}+\varepsilon_{i}$$
(4)$$socio_{i}=\alpha_{4}+\beta_{7}parenting1st_{i}+\beta_{8}gender_{i}+\beta_{9}parenting1st_{i}\times gender_{i}\;+\beta_{10}parenting2nd_{i}+\beta_{11}gender_{i}\times parenting2nd_{i}\;+\gamma X_{i}+u_{j}+\varepsilon_{i}$$
where $gender_{i}$ is G3/G4’s gender, ceteris paribus. Equation (3) examines the first stage effect, or the regression coefficient, of the path from the parenting styles of G1 to G2. Equation (4) examines the second stage effect, namely, the regression coefficient of the path from the parenting styles of G2 to the social–emotional development of G3/G4. Additionally, it also includes the direct effect of the parenting styles of G1 on the social–emotional development of G3/G4. The indirect effect can be calculated by multiplying the first and second-stage effect coefficients. We also tested the confidence interval of these effects using the bootstrap method based on a resampling with 1000 replications. This was performed in order to validate the coefficients and significance of each effect.

Several conditions should be held in order to claim the existence of the moderated mediation effect. First, the coefficient of the indirect effect of treatment on the mediator—that is, the $\beta_{4}$ in Equation (4)—should be statistically significant. Second, in Equation (4), the coefficient of the effect of the interaction term between the mediator and the moderator ($gender_{i}\times parenting2nd_{i}$) on the outcome ($\beta_{11}$) should be significant; the coefficient of the effect of the interaction term between the independent variable and the moderator ($parenting1st_{i}\times gender_{i}$) on the outcome ($\beta_{9}$) should be significant; or the coefficient of the effect of the mediator on the outcome ($\beta_{10}$) should be statistically significant.

## 5. Results

### 5.1. Intergenerational Transmission of Parenting Styles

In Table 1, we report the descriptive statistics for our sample. Regarding socioeconomic characteristics, about half (51.03%) of the G3/G4 were male; these children were slightly over 11 months old on average at the time of data collection. Furthermore, the primary caregiver G2’s average educational attainment was the middle school stage. G3/G4 possessed 0.49 siblings at home, on average. Regarding parenting styles, the average score for the emotional warmth and rejection parenting styles of G1 were 3.93 and 4.92, respectively. The average score for the warm and hostile parenting styles of G2 were 3.44 and 1.62, respectively. Regarding social–emotional problems and the competencies of G3/G4, the average score for social–emotional problems was 6.76; additionally, the average score for social–emotional competencies was 4.01.

### 5.2. The Mediation Effects of G2’s Parenting Styles (Hypothesis 1a, Hypothesis 1b, and Hypothesis 2)

In order to test our hypotheses on the total influence of the G1 parenting styles, the total standardized effects of the parenting styles of G1 on that of G2, as well as the socio-emotional problems and competencies of G3/G4, were estimated. The results in Figure 2 show that, in line with Hypothesis 1, the emotional warmth and rejection parenting styles of G1 were positively associated with the warm and hostile parenting styles of G2, respectively. In addition, the intergenerational transmission of warm parenting styles (*β* = 0.665, *p* < 0.01) was larger than that of the rejection and hostile parenting styles (*β* = 0.619, *p* < 0.01).

The results of the mediation analysis also supported our second hypothesis, which postulated that the warm parenting styles of G2 are positively associated with the social–emotional competencies of G3/G4 (*β* = 0.072, *p* < 0.05). Furthermore, they are also negatively related to the social–emotional problems of G3/G4 (*β* = −0.114, *p* < 0.1), whereas the hostile parenting styles of G2 are positively associated with the social–emotional problems of G3/G4 (*β* = 0.093, *p* < 0.05) and are negatively related to the social–emotional competencies of G3/G4 (*β* = −0.116, *p* < 0.05).

We found that the warm and hostile parenting style of G2 mediated the relationship between the emotional warmth and rejection parenting style of G1 and the social–emotional development of G3/G4—Hypothesis 3 was thus proved. The detailed regression that produced these results can be found in the Appendix A
Table A1.

The bootstrap results indicated that a 95% bias-corrected confidence interval did not contain zero (Table 2). Following the results in Figure 2, the results in Table 3 again proved that: for the estimated indirect effects of the parenting styles of G1 on the social–emotional competencies of G3/G4 through the warm parenting style of G2, the point estimates are significantly larger than zero. For instance, one standard deviation increase in the rejection parenting style of G1 is associated with a 0.074 SD decrease in the social–emotional competencies score of G3/G4 at the 5% significance level (see Row 8, Panel B, Table 2). This finding also strongly suggests that the indirect effects of the parenting style of G2 are statistically significant.

We also examined the different mediation effects of G2’s parenting styles on the association between the parenting styles of both male and female caregivers in G1, i.e., the grandfathers/great grandfathers and grandmothers/great grandmothers of G3/G4, respectively, as well as the social–emotional development of G3/G4. The results in Appendix A
Table A2 and Table A3 verified the abovementioned mediation effects. The results also suggested that the parenting styles of both the grandfathers/great grandfathers and grandmothers/great grandmothers were associated with the parenting outcomes of their offspring.

### 5.3. The Moderation Effect of the Gender of Great Grandchildren/Grandchildren (Hypothesis 3)

Having examined the mediation effect of the parenting styles of G2, we further analyzed the conditional indirect effect of the parenting styles of parents in G1 on the social–emotional development of G3/G4 (through the parenting styles of G2), understood through the gender of G3/G4. The preliminary results for the moderated mediation effects are shown in Table 3, Table 4, Table 5 and Table 6.

Consistent with previous findings, the rejection parenting style of grandfathers/great-grandfathers (*β* = −0.242, *p* < 0.1, Model 1, Table 4) negatively predicted the warm parenting styles of G2. Further, we found a significant moderating role of G3/G4’s gender in the first stage of the mediation, that is, from the parenting styles of G3/G4 to the parenting styles of G2 (*β* = 0.328, *p* < 0.05). In other words, when G3/G4 is a boy, G2 tends to be more prone to exert a warm parenting style, even if they perceived a rejection parenting style in childhood. Consequently, G3/G4 tends to have fewer social-emotional problems (*β* = −0.287, *p* < 0.05). We did not find other moderated mediation effects in the models (Table 5 and Table 6).

## 6. Discussion

In this study, we used new survey data collected from 194 rural households in three townships in rural China in order to extend the investigation into between-generation transmission effects of parenting styles and their association with the social–emotional development of grandchildren/great grandchildren. We also examined the roles of parent/grandparent’s parenting styles and grandchild/great grandchild’s gender behind the association.

In line with previous research, regarding the demonstration of the intergenerational transmission of parenting styles [11,12,13,14], this study also shows that both the warm and rejection parenting styles of G1 are significantly correlated with the corresponding parenting styles of G2. Our findings also suggest that the parenting styles of both the grandfathers/great grandfathers and grandmothers/great grandmothers are associated with the parenting outcomes of their offspring. Most studies agree that mothers play a more fundamental role in the child’s development than the fathers [53]. Historically, it has been thought that the father bears less responsibility for the care and training of their children, but are more protective [54,55]. However, the mother spends more time interacting with their children and are more authoritative than the father [55]. In this sense, this specific finding is intriguing and potentially highlights a more significant role in regard to the father when the father plays as the primary caregiver, which does make sense as many families have seen increased male engagement in recent years.

As is consistent with the growing body of research that demonstrates that parental rejection and punishment can result in mental disorders and problematic behaviors (such as compulsion, hyperactivity, aggression, and discipline violation) among children [16], through our study, we demonstrate that the emotional warmth parenting styles of the G1 measured in this study can significantly predict the social–emotional competencies of G3/G4. In addition, the rejection parenting styles of G1 can predict the social–emotional problems of G3/G4. More importantly, the mediation model has shown that the parenting styles of G2 mediate the associations. Specifically, caregivers who have experienced rejection parenting styles during childhood and adolescence are more likely to adopt a hostile parenting style when they themselves become caregivers. As such, there is a possibility that the caregivers’ hostile parenting style has a detrimental effect on the child’s social–emotional development.

On the other hand, parenting styles may have been an intermediate factor. Warm parenting styles of caregivers lead to better social–emotional conditions in children and may introduce fewer social–emotional problems among the generation after the next generation. A key finding from this study is that the consequences of less optimal parenting styles can be transgenerational and bitterly opposed to change.

The gender of the child matters when analyzing the mediating role of the parenting styles of the grandparent/parent. Parenting styles mediate the intergenerational association when G3/G4 is a boy but not when they are a girl. This finding suggests that when a boy whose grandparent/great grandparent exhibits a rejection parenting style toward the boy’s parent/grandparent, then the parent/grandparent tends to be caring and supportive toward the boy, which puts the boy at a lower risk of social–emotional problems.

This study contributes to the evidence base in several ways. Most earlier studies that suggested parenting styles predict a child’s social–emotional development were based on two-generation models. While some studies have examined the relationship between parenting and the developmental outcomes of the grandchild in the three-generation model, this type of research only typically examined zero-order associations. Further, almost all the studies in this area were conducted in Western cultures.

Due to factors such as culture and social background, families in rural China frequently overlook parent–child interaction because rural caregivers believe that “enough food and warm clothes” is sufficient to nurture children. The early development potential of rural children cannot be fully ensured as the caregivers lack positive parenting styles. In addition, as urbanization progresses, many parents migrate to work in large cities and leave their children in the care of grandparents with less optimal parenting styles. This study is the first study to investigate the intergenerational association between the parenting styles of grandparents/great-grandparents and the neurodevelopment of grandchildren/great-grandchildren in a relatively less developed region, i.e., rural China, so as to provide references for the government to provide service policies related to early childhood development and provide supplementary evidence for a more comprehensive understanding of early childhood development worldwide.

We believe that the results from this study, in conjunction with the findings of prior work, may provide additional implications in respect of the preventive interventions that break the vicious associations of poor parenting styles across generations. First, the mediating role of G2’s parenting styles in regard to the intergenerational association between the parenting styles of G1 and the social–emotional problems of G3/G4 suggests that parent-training interventions that aim to give a perspective on how parents should raise their children scientifically may be helpful in respect of interrupting the deterioration of social–emotional problems among children. Second, supposing the current parenting style status is constructed partly by the parents’ perception of how they were reared, the representation of the experience regarding the parents may be an essential target for the purposes of therapeutic intervention. Third, the development of female children deserves more attention. The patrilineal family systems and associated socio-cultural practices are innately biased toward sons. Daughters marry into their husbands’ families to carry on the family line, whereas adult males stay with their parents to care for their aging parents. In China, proverbs such as “a son keeps incense at the ancestral altar burning” and “investing in a girl is like pouring water onto another’s land” represent the various roles that sons and daughters play within the family structure. Consistent with this social phenomenon, this study found that girls receive less caring and emotional support from family members, and are more susceptible to the negative consequences of rejective parenting styles. Parenting interventions should include educational campaigns on gender equality, as the development of females is an important component of human capital in a society, which contributes to social and economic growth.

Several limitations should be acknowledged. First, the cross-sectional nature of our study did not allow us to infer cause–effect relationships. Second, the assessment of the parenting styles of G1 was retrospective, it cannot perfectly examine inter-temporal causality and is subject to recall bias. Additionally, the perceived parenting styles of G1, as self-reported by G2, may not reflect the actual parenting style of G1. In the future, when using panel data with prospective variables, it would be interesting to examine further the causal relations between the parenting styles of different generations and their effects on the neurodevelopment of young children. Third, we urge caution regarding generalizing these findings to other contexts, given that our sample was collected from a typical rural area in East China, and the sample size was relatively small. Although the sample size was adequate to meet the model requirements, the small sample size did not allow detailed subgroup analyses. It seems reasonable that the different intergenerational relationships may be observed in families with different cultural and socioeconomic backgrounds. Further studies should increase sample size in more diverse contexts to generate more in-depth and representative knowledge on the relationship between emotional warmth and rejection parenting styles of grandparents/great-grandparents and the social–emotional development of grandchildren/great-grandchildren, as well as explore heterogeneity across families with different cultural and socioeconomic backgrounds.

## 7. Conclusions

Based on a sample of 194 primary caregivers of children aged between 6 and 36 months from rural China, in this study, evidence was found that the warm and hostile parenting styles of parents/grandparents mediated the relationships between the emotional warmth and rejection parenting styles of grandparents/great grandparents and the socio-emotional development of their grandchildren/great grandchildren. When the child is a boy, parents/grandparents frequently employ a warm parenting style, which leads to less socio-emotional issues in respect to the child. As such, future studies with longitudinal research designs are needed in order to confirm the intergenerational effects discussed in this paper.

## Figures and Tables

**Figure 1 ijerph-20-01568-f001:**
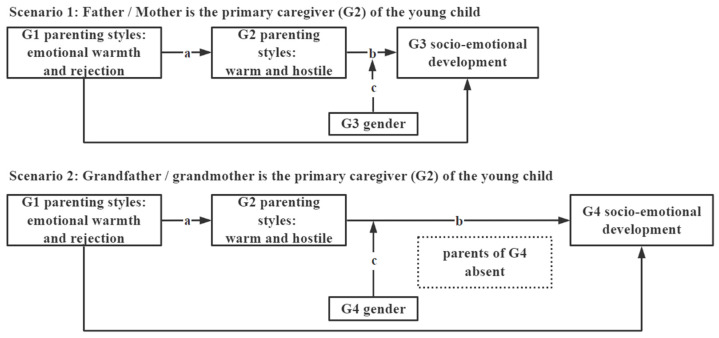
Intergenerational association model; dependent variable is the socio-emotional development of G3/G4. Note: a: the direct effects of parenting styles of G1 on parenting styles of G2. b: The direct effect of parenting styles of G2 on the socio-emotional development of G3/G4. The direct effects of parenting styles of G1 on the socio-emotional development of G3/G4 are non-mediated effects. c: the moderated mediation effect of G3/G4’s gender.

**Figure 2 ijerph-20-01568-f002:**
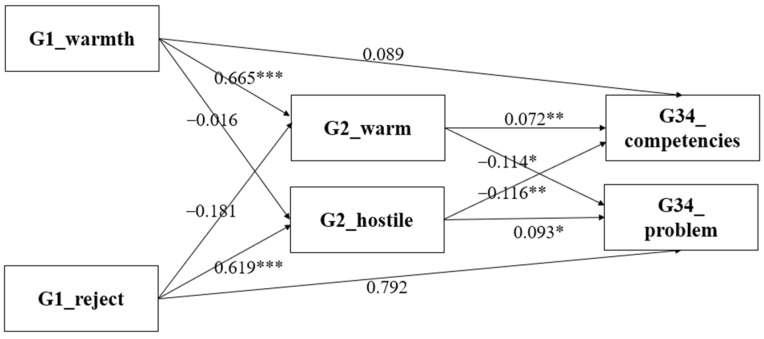
Intergenerational association model; dependent variable is the socio-emotional competencies and problems of young children. * *p* < 0.05; ** *p* < 0.01; *** *p* < 0.001

**Table 1 ijerph-20-01568-t001:** Descriptive statistics.

Variable	Definition	Mean ± S. D/No. (%)
Dependent variable		
G3/G4_problem	The score of children’s social-emotional problems	6.76 ± 4.04
G3/G4_competencies	The score of children’s social-emotional competencies	4.01 ± 2.1
Independent variable		
G1_warmth G1_reject	The score of perceived emotional warmth experience with parents The score of perceived rejection experience with parents	3.93 ± 6.38 4.92 ± 6.15
Mediator variable		
G2_warm	The score of warm parenting style of G2	3.44 ± 0.52
G2_hostile	The score of hostile parenting style of G2	1.62 ± 0.46
Covariates		
Gender	G3/G4’s gender	
Boy		99 (51.03)
Girl		95 (48.97)
Age in months	Age of G3/G4’s in month	11.32 ± 4.72
Male siblings	Number of male siblings	0.15 ± 0.37
Child number	Number of siblings in total	0.49 ± 0.73
Low birth weight	Whether G3/G4 was born with low birth weight	
Yes		44 (22.68)
No		150 (77.32)
Caregiver’s age	Caregiver’s age in years	38.22 ± 13.67
Caregiver’s education	Caregiver’s educational attainment	3.23 ± 1.17
Caregiver’s Generation	Whether G2 is the parent of G3 or the grandparent of G4	
father/mother		118 (60.82)
grandfather/grandmother		76 (39.18)
Dibao	Whether the family is receiving subsistence allowances	
Yes		32 (16.49)
No		162 (83.51)

Data source: Authors’ survey. N = 194.

**Table 2 ijerph-20-01568-t002:** Bootstrap estimates of indirect effects of the parenting style of G1 on social–emotional developments (problems and competencies) of G3/G4 through the parenting styles of G2.

Indirect Effect		Point Estimate	Bootstrap S.E.	95% CI (Percentile)	95% CI (B.C.)
		(1)	(2)	(3)	(4)
	Panel A				
(1)	G1_warmth on G3/G4_ problems through				
	G2_warm	0.098	0.072	[0.011, 0.073]	[0.004, 0.081]
(2)	G1_reject on G3/G4_ problems through				
	G2_warm	−0.035	0.047	[−0.138, 0.056]	[−0.221, 0.02]
(3)	G1_warmth on G3/G4_ competencies through				
	G2_warm	0.052	0.029	[0.006, 0.122]	[0.013, 0.132]
(4)	G1_warmth on G3/G4_ competencies through				
	G2_warm	−0.018	0.024	[−0.073, 0.023]	[−0.084, 0.015]
	Panel B				
(5)	G1_warmth on G3/G4_ problems through				
	G2_hostile	−0.001	0.037	[−0.077, 0.084]	[−0.086, 0.071]
(6)	G1_reject on G3/G4_ problems through				
	G2_hostile	0.052	0.067	[−0.079, 0.187]	[−0.066, 0.196]
(7)	G1_warmth on G3/G4_ competencies through				
	G2_hostile	0.001	0.026	[−0.049, 0.061]	[−0.049, 0.061]
(8)	G1_reject on G3/G4_ competencies through				
	G2_hostile	−0.074	0.043	[−0.18, −0.011]	[−0.174, −0.009]

Notes: (i) The mediator is the score of G2’s parenting styles. Panel A represents the indirect effects of the parenting style of grandparent/great-grandparent on social–emotional developments (problems and competencies) of great-grandchild/grandchild through warm parenting styles of parent/grandparent. Panel B represents the above indirect effects through hostile parenting styles of parent/grandparent. (ii) Bootstrap standard errors are based on resampling with 1000 replications. (iii) The percentile 95% CI uses usual sampling distribution cutoffs without bias correction, while the BC 95% CI corrects for a bias in the distribution of bootstrap estimates.

**Table 3 ijerph-20-01568-t003:** Testing the moderated mediation effects of G3/G4’s gender when the predictors are the emotional warmth parenting style of G1 parents and the mediator is G2’s emotional warmth parenting style.

Predictors	G2_ Warm	G3/G4_ Problem	G3/G4_ Competencies	G2_ Warm	G3/G4_ Problem	G3/G4_ Competencies
	(Model 1: X = G1_Warmth_Dad)			(Model 2: X = G1_Warmth_Mom)		
X	0.302 **	−0.036	0.016	0.253 **	−0.063	0.003
	(0.084)	(0.071)	(0.079)	(0.073)	(0.099)	(0.076)
MO: G3/G4_ gender	0.136	−0.033	−0.071	0.141	−0.034	−0.070
	(0.116)	(0.111)	(0.122)	(0.116)	(0.110)	(0.124)
XMO	−0.150	0.031	0.108	−0.067	0.093	0.053
	(0.160)	(0.101)	(0.102)	(0.135)	(0.144)	(0.127)
ME: G2_ warm		0.063	0.165 *		0.070	0.162 *
		(0.125)	(0.076)		(0.124)	(0.076)
MEMO		0.059	−0.086		0.046	−0.077
		(0.141)	(0.128)		(0.139)	(0.132)
R^2^	0.115	0.260	0.461	0.112	0.261	0.457

Notes: (i) The covariates that are controlled in the regressions include the child’s age in months, birth order, number of siblings at home, whether the child was born with low birth weight, caregiver’s age in years, and education level, whether the G2 caregiver is the mother/father of G3 or the grandmother/father of G4, whether the family is receiving subsistence allowances. Standard errors present in parentheses are clustered at the village level. * *p* < 0.05; ** *p* < 0.01.

**Table 4 ijerph-20-01568-t004:** Testing the moderated mediation effects of G3/G4’s gender when the predictors are the rejection parenting style of G1 parents and the mediator is G2’s emotional warmth parenting style.

Predictors	G2_ Warm	G3/G4_ Problem	G3/G4_ Competencies	G2_ Warm	G3/G4_ Problem	G3/G4_ Competencies
	(Model 1: X = G1_Reject_Dad)			(Model 2: X = G1_Reject_Mom)		
X	−0.242 †	0.357 **	−0.126 †	−0.121	0.348 ***	−0.128 †
	(0.132)	(0.095)	(0.073)	(0.135)	(0.095)	(0.069)
MO: G3/G4_gender	0.124	0.001	−0.066	0.139	−0.000	−0.068
	(0.108)	(0.111)	(0.123)	(0.113)	(0.109)	(0.124)
XMO	0.328 *	−0.287 *	−0.176	0.224	−0.179	−0.188
	(0.153)	(0.134)	(0.093)	(0.132)	(0.122)	(0.095)
ME: G2_warm		0.158	0.197 *		0.106	0.180 *
		(0.106)	(0.074)		(0.096)	(0.072)
MEMO		−0.042	−0.099		−0.004	−0.080
		(0.118)	(0.125)		(0.106)	(0.123)
R^2^	0.099	0.320	0.464	0.079	0.331	0.465

Notes: (i) The covariates that are controlled in the regressions include the child’s age in months, birth order, number of siblings at home, whether the child was born with low birth weight, caregiver’s age in years, and education level, whether the G2 caregiver is the mother/father of G3 or the grandmother/father of G4, whether the family is receiving subsistence allowances. Standard errors present in parentheses are clustered at the village level. † *p* < 0.10; * *p* < 0.05; ** *p* < 0.01; *** *p* < 0.001.

**Table 5 ijerph-20-01568-t005:** Testing the moderated mediation effects of G3/G4’s gender when the predictors are the emotional warmth parenting style of G1 parents and the mediator is G2’s hostile parenting style.

Predictors	G2_ Hostile	G3/G4_ Problem	G3/G4_ Competencies	G2_ Hostile	G3/G4_ Problem	G3/G4_ Competencies
	(Model 1: X = G1_Warmth_Dad)			(Model 2: X = G1_Warmth_Mom)		
X	0.005	−0.016	0.036	−0.041	−0.042	0.028
	(0.157)	(0.070)	(0.075)	(0.133)	(0.094)	(0.079)
MO: G3/G4_gender	0.299 *	−0.059	−0.007	0.299 *	−0.059	−0.005
	(0.147)	(0.124)	(0.119)	(0.147)	(0.123)	(0.121)
XMO	0.011	0.028	0.069	0.062	0.090	0.039
	(0.160)	(0.097)	(0.098)	(0.142)	(0.140)	(0.119)
ME: G2_hostile		0.094	−0.226 *		0.092	−0.223 *
		(0.103)	(0.092)		(0.105)	(0.092)
MEMO		0.060	0.160		0.061	0.156
		(0.137)	(0.125)		(0.137)	(0.124)
R^2^	0.124	0.264	0.473	0.124	0.266	0.469

Notes: (i) The covariates that are controlled in the regressions include the child’s age in months, birth order, number of siblings at home, whether the child was born with low birth weight, caregiver’s age in years, and education level, whether the G2 caregiver is the mother/father of G3 or the grandmother/father of G4; whether the family is receiving subsistence allowances. Standard errors present in parentheses are clustered at the village level. * *p* < 0.05.

**Table 6 ijerph-20-01568-t006:** Testing the moderated mediation effects of G3/G4’s gender when the predictors are the rejection parenting style of G1 parents and the mediator is G2’s hostile parenting style.

Predictors	G2_ Hostile	G3/G4_ Problem	G3/G4_ Competencies	G2_ Hostile	G3/G4_ Problem	G3/G4_ Competencies
	(Model 3: X = G1_Reject_Dad)			(Model 4: X = G1_Reject_Mom)		
X	0.151 †	0.311 **	0.112	0.182 *	0.330 ***	−0.151 *
	(0.083)	(0.088)	(0.074)	(0.095)	(0.096)	(0.082)
MO: G3/G4_gender	0.340 *	−0.012	0.006	0.333 *	−0.008	0.006
	(0.137)	(0.127)	(0.122)	(0.150)	(0.125)	(0.122)
XMO	0.088	−0.263 †	−0.139	0.091	−0.178	−0.186
	(0.130)	(0.135)	(0.105)	(0.100)	(0.121)	(0.110)
ME: G2_hostile		0.047	−0.240 *		0.032	−0.252 *
		(0.102)	(0.092)		(0.109)	(0.096)
MEMO		0.089	0.180		0.071	0.196
		(0.124)	(0.128)		(0.133)	(0.131)
R^2^	0.162	0.312	0.472	0.176	0.325	0.477

Notes: (i) The covariates that are controlled in the regressions include the child’s age in months, birth order, number of siblings at home, whether the child was born with low birth weight, caregiver’s age in years, and education level, whether the G2 caregiver is the mother/father of G3 or the grandmother/father of G4, whether the family is receiving subsistence allowances. Standard errors present in parentheses are clustered at the village level. † *p* < 0.10; * *p* < 0.05; ** *p* < 0.01; *** *p* < 0.001.

## Data Availability

The data presented in this study are available on request from the corresponding author. The data are not publicly available due to it contains sensitive information of children and parents in rural China.

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
