# Peer review of "Emotional Warmth and Rejection Parenting Styles of Grandparents/Great Grandparents and the Social–Emotional Development of Grandchildren/Great Grandchildren"

_ijerph, 2023, doi:10.3390/ijerph20021568_

Round 1
Reviewer 1 Report
The study is on an interesting and important area and conducted among a high-risk population which consists of rural children and parents who have low access to resources in China. The study focused on emotional warmth and rejection in parenting styles and their intergenerational effects on grandchildren’s socio emotional development.
The overall goals are praiseworthy, there are however some areas that need further explanation and clarification. Overall, in the introduction, citations for some for overall rationale for the study are not clearly mentioned, some terminology is incorrect, and the overall introduction would need to be strengthened to make a clearer and more compelling case for the study. There are also clarifications needed for the methodology as well as the interpretation of results.
Overall, the manuscript requires a major revision and substantial copy editing. First, the text is difficult to follow, in part, because of poor translation to English. I recognize that translation of such writing is difficult and requires substantial time and effort. Nevertheless, there are many errors in grammar and language use throughout the manuscript. These errors substantially interfered with understanding many of the key points and limit both readability and the potential of the manuscript to contribute to the literature.
In addition to translation, revision, and editing for clarity, the results section is presented in a fashion that makes it very difficult to understand and interpret results. I encourage the authors to rework this section. First, less detail will lead to greater ability to communicate the key findings to readers. As currently written, the key findings and their importance is lost to the volume of detail and the formatting of their presentation. One possible way to clarify is to present the primary models using figures of the path models. Readers are likely to more quickly and readily grasp associations among the variables from such figures, to understand the overall model, and to identify both significance and strength of associations of the variables. Needed additional detail can be conveyed if they are displayed in tables, appendices, or other vehicles.
Introduction:
Some specific areas to clarify in revision: In the first paragraph of the introduction, the authors mention how parenting style reveals something about “human capital and inequality,” without defining what they are considering human capital and inequality to be in this context. Next in the third paragraph, authors note that “parenting styles influence the human capital formation, especially during the early stages in life”: in this part of the introduction, there needs to be additional information from a literature review on human capital formation since it was mentioned multiple times without being defined. Additionally, given the focus of the study, these constructs need to be applied to developmental trajectories across time and on how it might shift and change.
In Figure 1, it was very hard to follow the different variables as each variable was not described underneath the figure. We suggest that they add a description of variables in addition to an explanation of the models. For instance, there is an arrow coming from G1 to G4 that we suggest they explain further and tie it to the main question. Further, authors may consider moving figures until after they have introduced the hypotheses to visually represent them.
In the theoretical background and hypotheses section, “studies conducted in Pakistan and Spanish” need to be changed to “Pakistan and Spain”. Overall, the introduction provided a general idea to lay the theoretical grounds to the relationships between parenting styles and child outcomes, yet there was not a theoretical framework described to explain the specific hypotheses that the researchers decided to test. Authors may consider a more extensive literature review detailing: why do they think gender would be a moderator? Authors did cite a few studies showing that gender is indeed an effect but there was no theoretical rational explaining it. Authors may also consider addressing what cultural factors may explain this hypothesis.
Methodology:
The authors are testing mediation and moderated mediation models in which they argue support for generational transmission (a temporal order) using cross-sectional data. In this case, the paths can arguably be in either direction since the design is not longitudinal. Although theoretically the authors are measuring events that occurred in the past (e.g. parenting styles of great grandparents), this study does not directly test temporal ordering based on the measurement strategy as it is not a prospective study.
At a minimum, it is standard to test a reverse causality model when using cross-sectional data to test mediation. If associations and model fit are unchanged when testing such a model, direction of effects cannot be inferred. If model fit to data declines when testing reverse causality, inference about the proposed direction of effects is strengthened. There are additional considerations for inference in this case. See, e.g.,
S. E., D. A. and M. A. (2011). Bias in cross-sectional analyses of longitudinal mediation: Partial and complete mediation under an autoregressive model. Multivariate Behavioral Research, 46: 816–841.
Shrout PE. (2011). Commentary: Mediation Analysis, Causal Process, and Cross-Sectional Data. Multivariate Behavioral Research, 46: 852-60.
It is not clear how many dropouts or missing scores there are. Authors indicated that they ended up with 194 but did not provide sufficient information on missing data. The authors indicated use of FIML to handle missing data. Validity of estimates using FIML depend on whether data can be characterized as missing at random.
Authors mention “ first study to examine correlation,” we suggest changing the word correlation to “association” since the authors used mediation models.
Additionally, authors used questionnaires to assess warmth and hostile parenting styles, but their literature review discussed papers with authoritative, authoritarian, or other parenting styles. It seems that some scales were chosen from the measures. The manuscript would benefit from providing a detailed explanation of what is considered a warm or hostile parenting style in the very beginning of the introduction as it clearly sets the definitions of variables from the beginning. Additionally, please clarify scoring of parent variables in terms of the combined scores and difference scores. The process for constructing these variables is unclear.
Parenting variables created for this study seem to differ in scoring from those previously used for the parenting measures. Were psychometric properties of the current variables tested?
Page 5, lines 199-200 should read “…factorial and construct validity…” (not constructed validity)
In describing analyses there are several errors. First, the authors stated that they “centralized” variables. I believe they mean to state “mean centered,” which is standard terminology.
In section 4.2 describing the moderated mediation model, the text refers to “…mediation effect and mediation effect.” I believe the authors intended to describe mediation and moderation effects here.
Author Response
Attached below please see the response letter for Reviewer1.

Reviewer 2 Report
This paper deals with a complex and interesting topic. It provides scientific evidence on the subject of parental styles in grandparents and great-grandparents and socioemotional development in grandchildren and great-grandchildren. Therefore, a three-generation model is considered. It is contextualized in a rural area of China and uses a sample of nearly 200 primary caregivers of children aged 6-36 months.
The theoretical framework adequately shows the state of the art, justifying the relevance of the study and setting out the study hypotheses. Similarly, the material and methods section describes the fundamental elements that make up the study, such as the participants, the measurement instruments, the data collection process and the details of the analyses carried out.
The results are comprehensive and are correctly presented and structured in various tables, responding to the working hypotheses. The research corroborated the existence of intergenerational transmission of parenting styles; both warm and hostile styles in G1 were significantly correlated with the corresponding parenting styles in G2. It has also been shown the relationship between warm/unfriendly parenting styles with social-emotional competencies/problems and that styles may be transgenerational, opposing as the authors say "bitterly to change". There is a mediating role of G2 parenting styles in the intergenerational association between G1 parenting styles and children's socioemotional behavior. This is important to take into account for pedagogical and therapeutic interventions. It would be very interesting for future work to address these interventions and observe positive results in this aspect.
In terms of gender, there were also noteworthy findings. Thus, the father played the role of main caregiver in an important way, which implies progress in this sense. On the other hand, when the child is male, the father/grandfather frequently employs a warm parenting style and this leads to fewer socioemotional problems. In this sense, it seems necessary to continue breaking stereotypes and re-educating so that girls are not placed in more disadvantaged positions and have an equitable education. Along the same lines as highlighted above, it seems urgent to implement educational measures from a gender perspective and focused especially on girls.
Perhaps, future lines of research could be explained in more detail, based on the discussion made in the document. In any case, this is a solid, well-written document with educational and scientific relevance. I recommend that it be published and that in the future interventions can be carried out to improve the socioemotional competencies of everyone in general and of girls in particular.
There are some formal aspects of the journal that need to be corrected, which I understand will be corrected in the final phase of the process.
Author Response
Attached below please see the response letter for Reviewer2.
